# Immunoregulatory Roles of Osteopontin in Diseases

**DOI:** 10.3390/nu16020312

**Published:** 2024-01-20

**Authors:** Lebei Wang, Xiaoyin Niu

**Affiliations:** 1Department of Immunology and Microbiology, Shanghai Institute of Immunology, Shanghai Jiao Tong University School of Medicine, Shanghai 200025, China; gloria_wlb@sjtu.edu.cn; 2College of Stomatology, Shanghai Jiao Tong University School of Medicine, Shanghai 200025, China

**Keywords:** osteopontin, immunoregulation, diseases

## Abstract

Osteopontin (OPN) is a multifunctional protein that plays a pivotal role in the immune system. It is involved in various biological processes, including cell adhesion, migration and survival. The study of the immunomodulatory effects of OPN is of paramount importance due to its potential therapeutic applications. A comprehensive understanding of how OPN regulates the immune response could pave the way for the development of novel treatments for a multitude of diseases, including autoimmune disorders, infectious diseases and cancer. Therefore, in the following paper, we provide a systematic overview of OPN and its immunoregulatory roles in various diseases, laying the foundation for the development of OPN-based therapies in the future.

## 1. Introduction

Osteopontin (OPN), also known as bone sialoprotein I (BSP-1), early T-lymphocyte activation (ETA-1) and secreted phosphoprotein 1 (SPP1), is a glycoprotein which is detected naturally in the bones, kidneys and lungs and is, furthermore, expressed by various immune cells, including T cells, B cells, natural killer (NK) cells, dendritic cells (DCs) and macrophages [1,2]. As a growth regulatory protein, multifunctional cytokine, and adhesion molecule, OPN is involved in the pathological processes of many diseases. The highest concentration of OPN in the human body is found in colostrum, where it acts as a modulator of the intestinal immune homeostasis and gut microbiome in newborns and infants [2]. OPN participates in a wide array of biological functions, including the stimulation and modulation of the immune system, the process of biomineralization, tissue remodeling activities and bacterial interactions [3]. However, the regulation of OPN expression may differ among various cell types and is not fully understood yet. Here, we provide a systematic overview of OPN and its immunoregulatory roles in various diseases, highlighting the potential of OPN-based therapies.

## 2. Structure of OPN

OPN is a highly phosphorylated glycoprotein which is rich in aspartic acid and has acidic characteristics consisting of 300 amino acids and including *O*-linked and *N*-linked oligosaccharides [4]. The gene structure of OPN includes a signal peptide sequence and a mature peptide sequence, the latter of which can be selectively spliced to produce different variants. The molecule of OPN undergoes several post-translational modifications, including phosphorylation and glycosylation. In the human body, there are three splice variants, namely OPN-a, OPN-b and OPN-c [5,6,7,8,9].

All three variants of OPN encompass several preserved components, including a sequence rich in aspartic acid at the *N*-terminal, and a calcium-binding site, a heparin-binding domain and a CD44-binding site at the *C*-terminal. The central part of OPN has a primary cluster of integrin-binding sites, comprising an Arg-Gly-Asp (RGD) motif and a SVVYGLR domain, followed by a thrombin cleavage site. RGD is associated with various integrins, such as αvβ1, αvβ3 αvβ5, αvβ6 and α5β1, while SLAYGLR revealed by thrombin cleavage attaches to α9β1, α4β1 and α4β7 (Figure 1) [3,10,11,12]. Moreover, thrombin, matrix metalloproteinases (MMPs), caspase-8/3, plasmin, cathepsin D and enterokinase are recognized as proteases that sever OPN at various locations, leading to the creation of multiple fragments, which include *N*-terminal fragment (*N*-OPN), *C*-terminal fragment (*C*-OPN), OPN-Arg168 (OPN-R), OPN-Leu167 (OPN-L) etc., with diverse sizes and functions [13,14].

## 3. Functions of OPN

OPN is a multifaceted protein which can exhibit diverse functions in different tissues (Table 1), interact with multiple receptors and regulate a variety of signaling pathways.

Resulting from alternative splicing and post-translational modifications, OPN has various isoforms that engage in multiple signaling pathways. The binding of OPN with integrins initiates the activation of several downstream signaling effectors, such as phosphatidylinositol 3 kinase (PI3K) /protein kinase B (AKT), focal adhesion kinase (FAK)/AKT and nuclear factor kappa-B (NF-κB), leading to cell proliferation, migration, epithelial–mesenchymal transition (EMT), inflammation, neurotoxic microglial phenotype, tumor growth, migration and invasion, as well as angiogenesis within the chronic subdural hematoma (CSDH) outer membrane [15,16,17,18,19,20]. In addition, OPN regulates other signal pathways or signal molecules, such as Janus kinase (JAK)/signal transducer and activator of transcription (STAT) [21,22,23,24], PI3K/AKT [15,25,26,27], NOTCH [28,29], extracellular regulated protein kinase1/2 (ERK1/2) [30], the ubiquitin *C*-terminal hydrolase L1 (UCHL1)–ubiquitin–proteasome system (UPS) axis [31] and transforming growth factor β (TGF-β) [32], influencing cellular physiological processes and disease progression. OPN also acts as a ligand for CD44 that results in attracting mesenchymal stem cells (MSCs) to the tumor microenvironment, promoting EMT and tumor budding (TB), inducing macrophage migration and activation, stimulating intestinal growth, differentiation and maturation, cell growth, proliferation, migration and cell-cycle activity and promoting oxidative stress [15,17,23,33,34,35,36].

**Table 1 nutrients-16-00312-t001:** Functions of OPN.

Functions	References
Cell proliferation	[15,16,19]
Cell migration	[15,19,34,36]
Cell-cycle activity	[35]
EMT	[15]
Tumor growth	[19]
Tumor migration and invasion	[19,23]
Angiogenesis	[20]
Inflammation	[24]
Oxidative stress	[36]

EMT: epithelial–mesenchymal transition.

## 4. Regulatory Effects of OPN on Immune Cells

OPN is expressed by macrophages in multiple pathologies and has been implicated in various functions. OPN regulates cytokine expression, such as interleukin-12 (IL-12) and IL-10, via OPN–αvβ3 and OPN–CD44 interactions. Inducible nitric oxide synthase (iNOS/NOS2) expression is also influenced by OPN as it reduces the stability of STAT1 that can bind to the NOS2 promoter. The absence of OPN results in defective ROS production and opsonization, thus affecting macrophage phagocytosis. As a small integrin-binding protein, OPN is also involved in the regulation of macrophage migration (Table 2) [37]. 

In addition, OPN is important for the maintenance of functional NK cell expansion. OPN blockade suppresses NK cell maturation and differentiation and induces NK cell apoptosis [38]. The absence of OPN in the microenvironment reduces the number of NK cells [39].

OPN is essential for the generation and differentiation of DCs. An intracellular form of OPN (iOPN) is a critical regulator for toll-like receptor-9 (TLR-9) and/or TLR-7-dependent interferon-α (IFN-α) expression by plasmacytoid DCs (pDCs) [40]. 

OPN plays an important role in innate immunity of allergic rhinitis (AR) by modulating group II innate lymphoid cells (ILC2s) and the interactions between ILC2s and eosinophils. In addition, OPN can promote ILC2 proliferation and upregulate the expression of GATA-binding protein 3 (GATA3), retinoid-related orphan receptor alpha (RORα), IL-5 and IL-13 [41].

OPN regulates the activation and aggregation of B cells. In vitro, recombinant osteopontin (rOPN) downregulates the co-stimulatory molecules CD80 and CD86 on B cells and reduces IL-6 production. However, rOPN also promotes B cell aggregation [42].

OPN plays significant roles in the generation, maturation, differentiation and activation of T cells. A secreted form of OPN (sOPN) is involved in the generation of T helper type 1 (Th1) and Th17 cells that are pathogenic T cells for various autoimmune diseases, while iOPN is necessary for Th17 cell development [40]. OPN modulates the generation of memory precursor effector CD8+ T cells by regulating cytokine milieu during the acute phase of virus infection [43] Moreover, Klement et al. demonstrated that downregulation of IRF8, a molecular determinant of apoptotic resistance in tumor cells and in CD11b + Ly6CloLy6G+ myeloid cells, aborted the repression of OPN. Thus, OPN can bind to its physiological receptor CD44 on activated T cells, acting as a potent T cell suppressor [44]. Similarly, myeloid and tumor cell–expressed OPN acts as an immune checkpoint to suppress T cell activation and confer host tumor immune tolerance [45]. In addition, mesenchymal stromal cells (MSCs) exert immunosuppressive effects on different immune cells. Resting MSCs can promote OPN production mediated by β1 integrin (CD29), whereas OPN expression is inhibited by prostaglandin E2 (PGE2) when MSCs are activated by proinflammatory cytokines [46].

**Table 2 nutrients-16-00312-t002:** Regulatory effects of OPN on immune cells.

	Immune Cells	Regulatory Effects	References
Stimulatory regulation	Macrophage	Promote cytokine and iNOS expression, phagocytosis and cell migration	[37]
	DC	Participate in the generation, development, differentiation and activation of DCs and regulate the expression of TLR-9 and IFN-α	[40]
	NK cell	Facilitate the maturation and differentiation of NK cells	[39]
	ILC	Enhance ILC2 proliferation and upregulate the expression of GATA3, RORα, IL-5 and IL-13	[41]
	T cell	Modulate the generation of Th1, Th17 and CD8+ T cells	[40,43]
	B cell	Increase the tendency of B cell aggregation	[42]
Inhibitory regulation	T cell	Suppress T cell activation	[44,45]
	B cell	Downregulate CD80 and CD86 on B cells, reduce the production of IL-6	[42]

iNOS: inducible nitric oxide synthase, DC: dendritic cell, TLR-9: toll like receptor-9, IFN-α: interferon-α, NK: natural killer, ILC: innate lymphoid cell, ILC2: group II innate lymphoid cell, GATA3: GATA binding protein 3, RORα: retinoid-related orphan receptor alpha, IL-5: interleukin-5, IL-13: interleukin--13, Th1: T helper type 1, Th17: T helper type 17, IL-6: interleukin-6.

## 5. Immunoregulatory Roles of OPN in Diseases

### 5.1. Digestive System Diseases

It is proposed that the activation of hepatic macrophages, particularly Kupffer cells (KCs), has significant roles in the development of metabolic-associated fatty liver disease (MAFLD) which encompasses a range of disease conditions from simple steatosis to non-alcoholic steatohepatitis (NASH) [47]. OPN has recently been described as a good biomarker of NASH [48]. In MAFLD, the number of KCs decreases and is supplanted by macrophages originating from the bone marrow. The recruited macrophages are divided into two subsets that either resemble homeostatic KCs or lipid-associated macrophages. The expression of OPN distinguishes the latter subset of recruited macrophages in the fatty liver. This provides an insight into macrophage-targeting strategies in MAFLD [49]. 

In NASH, OPN regulates macrophages and promotes liver injury through activation of various signaling pathways [24,34,50]. It promotes macrophage M1 polarization by activating the JAK1/STAT1/HMGB1 signaling pathway in hepatocytes, resulting in the increased expression of proinflammatory cytokines and liver injury [24]. Furthermore, sOPN derived from lipid-injured hepatocytes induces macrophage migration and activation via binding to CD44 receptor and activating the phosphorylation of the FAK (pFak)–NFκB signaling pathway. When silencing sOPN expression in hepatocytes or inhibiting sOPN release in NASH, macrophage infiltration, inflammation and fibrosis in the liver were reduced [34]. Moreover, intrahepatic OPN signaling by CREBZFOPN stimulates the activation of HSCs and fibrogenic cells and induces liver fibrosis and inflammation, worsening NASH severity [50]. Coiled-coil-helix-coiled-coil-helix domain-containing 2 (CHCHD2) is upregulated via YAP/TAZ-TEAD in NASH livers and consequently promotes liver fibrosis by activating the NOTCH pathway and enhancing OPN production [51]. These findings provide new insights into potential targets for NASH treatment, suggesting that targeting OPN or its downstream signaling pathway might be a novel therapeutic strategy. However, researchers found that colitis induced monocyte/macrophage infiltration in the gut and liver, promoting the expression of cholestasis-induced MoMF-Trem2 and Spp1, yet did not exacerbate liver fibrosis [52]. Han et al. concluded that macrophage-derived OPN protected hepatocytes from NASH by upregulating the oncostatin-M (OSM)-activated STAT3 signal and inducing arginase-2 (ARG2) and enhancing fatty acid oxidation (FAO) in hepatocytes [53]. 

Higher levels of plasma OPN are observed in inflammatory bowel disease (IBD) patients and are related to their clinical activity indices, and plasma OPN levels of Crohn’s disease (CD) patients are higher than those of ulcerative colitis (UC) patients [54]. Elevated OPN levels improve IL-12 production and are involved in the Th1 immune response related to CD [55]. OPN may also participate in the pathogenesis of UC [56], as the ratio of two types of large colon-infiltrating cells expressing OPN in the submucosa is higher than that in the CD group and diverticulitis group [57]. OPN haplotypes are modifiers of CD susceptibility, and the effects of OPN variants may regulate the secretion of IL-22 [58]. In intestinal inflammation induced by tumor necrosis factor-α (TNF-α), interferon regulatory factor 1 (IRF1) is activated and, thus, suppresses the expression of OPN, further inhibiting p-AKT, p-P38 and p-ERK activities and leading to aggravation of intestinal epithelial cell damage [59].

### 5.2. Urinary System Diseases

OPN might play a role in prostatic inflammation and fibrosis, which could lead to lower urinary tract symptoms (LUTS). The research conducted by Popovics et al. revealed that OPN protein levels were markedly elevated in the prostates of LUTS patients who underwent surgery, compared to incidental LUTS tissue. OPN secretion is triggered by proinflammatory cytokines, including IL-1β and transforming growth factor β1 (TGF-β1), and OPN directly influences stromal cells to promote the production of mRNA for proinflammatory cytokines. The pharmacological intervention of prostatic OPN could potentially alleviate LUTS by suppressing both the inflammatory and fibrotic pathways [60]. OPN is triggered by inflammation and extends the duration of the inflammatory condition. A genetic obstruction of OPN hastens the healing process following inflammation, which includes resolving prostate fibrosis [61]. The expression of OPN was negatively correlated with the expression of androgen receptor (AR), which is known to inhibit prostatic inflammation and fibrosis [62]. 

In chronic kidney disease (CKD), OPN may reflect disease risk and progression. Urinary OPN predicts incident CKD [63]. A population of profibrotic macrophages marked by expression of Spp1, Fn1 and Arg1 (termed Spp1 macrophages) expand in human chronic kidney disease. Platelet-instructed SPP1+ macrophages drive myofibroblast activation in fibrosis in a chemokine (C-X-C motif) ligand 4 (CXCL4)-dependent manner via Spp1, Fn1 and Sema3 crosstalk [64]. It is found that OPN is inversely related to estimated glomerular filtration rate (eGFR) and positively related to the urinary albumin–creatinine ratio (UACR). These associations are consistent across different subgroups of CKD patients. Furthermore, higher OPN levels are associated with a higher risk of kidney failure (KF) and all-cause mortality. *N*-OPN is carried by exosomes and secreted into the urine of patients with CKD and negatively correlated with kidney function [65]. Thus, OPN might be a potential biomarker and therapeutic target for CKD progression [66]. OPN, especially *N*-OPN, is encapsulated in β-catenin-controlled tubular cell-derived exosome cargo and subsequently passed to fibroblasts. Through binding with CD44, exosome OPN promotes fibroblast proliferation and activation [65].

In mice models of steroid hormone imbalance generated by the surgical implantation of testosterone (T) and estradiol (E2) pellets into male C57BL/6J mice, Popovics et al. found that steroid hormone imbalance increased macrophage infiltration, Spp1/OPN expression and lipid accumulation in the ventral prostate, leading to foam cell formation and urinary dysfunction. These effects were reduced or delayed in *OPN*-deficient mice, indicating that steroid hormone imbalance drove prostatic inflammation, fibrosis and proliferation, which were mediated by OPN secretion from luminal macrophages [67].

### 5.3. Hematological and Hematopoietic System Diseases

OPN plays a crucial role in inhibiting lymphoma development by regulating the STAT3 signaling pathway. Plasma and serum OPN levels are higher in myeloma, chronic myeloid leukemia and acute myeloid leukemia (AML) patients than those in healthy controls. It is a prognostic marker in AML, as high *OPN* mRNA expression levels suggest reduced event-free survival and overall survival [68]. The prognosis of diffuse large B cell lymphoma (DLBCL), a type of non-Hodgkin lymphoma, is related to OPN expression which is found to be associated with non-germinal center DLBCL, a more aggressive type of lymphoma [69].

Rizzello et al. observed that the deficiency of *OPN* increased the incidence and aggressiveness of splenic lymphomas in Faslpr/lpr mice, which resembled the activated type of DLBLC (ABC-DLBCL). This deficiency led to enhanced TLR9-MYD88 signaling in B cells, resulting in the activation of STAT3 and expression of myelocytomatosis viral oncogene homolog (c-MYC) and B cell lymphoma-2 (BCL2). Interestingly, it was found that the intracellular form of OPN (iOPN), but not the secreted form (sOPN), inhibited TLR9–MYD88–STAT3 signaling in B cell lymphoma cell lines, revealing that iOPN acted as a negative regulator of this pathway [21].

Furthermore, it is discovered that decreased autophagy inhibits OPN expression and downregulates JAK/STAT3 signaling, preventing lymphatic malformation from developing into lymphangiosarcoma. Ectopic expression of OPN in FIP200-null vascular tumor cells restores STAT3 phosphorylation and rescues their defective proliferation and tumorigenicity [22]. Autophagy inhibition prevents lymphatic malformation (LM) progression to lymphangiosarcoma (LAS) in vivo and reduces vascular tumor cell proliferation and tumorigenicity in vitro, without affecting mTORC1 signaling as an oncogenic driver [70].

### 5.4. Endocrine and Metabolic Diseases

Metabolic syndrome (MetS) is mainly caused by an imbalance between calorie intake and energy expenditure [71]. The cornerstones of MetS are determined as the four diseases of atherosclerosis, hypertension, obesity and diabetes [72]. Serum OPN levels may be an early biomarker to predict those four diseases significantly associated with MetS [29,36,73,74,75,76].

In atherosclerosis, gut dysbiosis increases the production of gut microbial lipopolysaccharide (LPS), which stimulates OPN expression in circulating monocytes and promotes vascular smooth muscle cell (VSMC) proliferation via the αvβ3/NF-κB pathway. Paeonol (Pae), a natural phenolic compound, inhibits atherosclerosis by reducing gut microbial LPS and OPN levels and blocking the crosstalk between monocytes and VSMCs [16]. Furthermore, OPN levels can strongly predict clinical outcome of stable patients with chronic heart failure [77].

OPN is associated with hypertension-related inflammatory cell recruitment and vascular remodeling via the AKT1/ activating protein-1 (AP-1) pathway [78]. Exogenous OPN may promote the differentiation of monocytes into an anti-inflammatory phenotype, reducing inflammatory cytokine expression, and may suppress macrophage-to-osteoclast development differentiation in hypertensive patients with vascular calcification (VC) [79,80]. In hypertensive patients, OPN is an independent risk factor for left ventricular (LV) hypertrophy and LV diastolic dysfunction. Nevertheless, OPN shows no association with LV dimension and systolic function [81].

In addition, OPN is involved in MetS due to obesity. Interventions such as exercise, diet and drugs can reduce the risk of metabolic syndrome and inflammatory response by inhibiting OPN expression. Exercise transiently decreases OPN which is involved in adipose tissue expansion and inflammation, in overweight individuals and individuals with obesity [82]. OPN could exacerbate high-fat diet (HFD)-induced metabolic dysfunctions by modulating the gut microbiome. *OPN* deficiency or OPN neutralization protects against HFD-induced lipid accumulation, liver damage and glucose intolerance. Further investigation shows that OPN decreases the adhesion of *Lactobacillus* to intestinal epithelial cells by inhibiting the expression of adhesion molecules via the NOTCH signaling pathway [29].

In diabetes, OPN levels are correlated with glycemic control [83]. *OPN* deficiency increases insulin sensitivity [84]. In adipose tissue, the accumulation of macrophage-derived OPN induces insulin resistance and triggers inflammation [85]. Moreover, OPN enhances the detection of low-grade inflammation in type 2 diabetes [86]. Type 1 diabetes is a disease that arises due to the autoimmune damage of insulin-producing pancreatic B cells [87]. In patients with type 1 diabetes, serum OPN levels are higher and are an essential predictor of incipient diabetic nephropathy and all-cause mortality [88].

In a mouse model of hypertension, wild-type mice showed an increase in reactive oxygen species production compared with *OPN* knockout mice [89]. In a diabetes mouse model, *OPN* knockout mice are protected from HFD-induced insulin resistance [90]. Neutralization of OPN decreases expression of hepatic gluconeogenic markers and increases apoptosis of macrophages in a diet-induced obese mouse model, reducing insulin resistance and obesity-associated inflammation [91].

### 5.5. Rheumatic Diseases

It is well known that inflammatory response is part of the pathogenesis of rheumatic diseases. Unlike classical extracellular matrix (ECM) proteins, OPN is a soluble protein and can induce cell motility and persistent inflammation rather than provide a scaffold for stable cell adhesion [40]. The upregulation of OPN during inflammation not only modulates the host response to infection but also promotes the development of immune-mediated inflammatory diseases, indicating its critical role in rheumatic diseases.

Rheumatoid arthritis (RA) is a common, inflammation-based disease characterized by abnormal immune cell infiltration in synovium, leading to the production of proinflammatory cytokines and limiting the patient’s movement [92,93,94]. Levels of OPN are closely related to IL-17 production as well as Th17 frequency in the synovial fluid of RA patients. The impact of OPN on Th17 differentiation is mediated by mechanisms that are independent of the IL-6/STAT-3 pathway or other cytokine-mediated mechanisms. In addition, it is found that OPN induces H3 acetylation of the *IL17A* gene promoter in CD4+ T cells primarily through the CD44-binding domain, thereby enabling the *IL17A* locus to interact with the transcription factor ROR [95]. IgG from RA patients with anti-cit-OPN antibodies increased the binding activity of OPN to fibroblast-like synoviocytes (FLSs), which further increased matrix metalloproteinase (MMP) and IL-6 production in TNF-stimulated FLSs. In patients with anti-cit-OPN antibodies, it is likely that B cells producing anti-cit-OPN aggravate arthritis [96]. Furthermore, thrombin activation of OPN and the subsequent inactivation by thrombin-activatable carboxypeptidase B (CPB) generate OPN-R and OPN-L, respectively. These processes play a central homeostatic role in RA by regulating neutrophil viability and reducing synoviocyte adhesion [14]. It is reported that OPN in plasma, synovial fluid and articular cartilage is associated with progressive joint damage and is probably a useful biomarker for determining the severity and progression of RA [97,98].

Osteoarthritis (OA) is a complex disease that mainly affects the joints and its pathogenesis is still not fully understood. The release of inflammatory mediators, such as IL-1β, IL-6 and TNF-α, and degradative enzymes from cartilage, subchondral bone and synovium may play a key role in disease pathogenesis [99]. OPN has a significant impact on the progression of OA. The expression levels of OPN are elevated in all OA cartilage compared with those in normal cartilage and are positively correlated with the Mankin score, indicating that they are associated with cartilage degeneration and chondrocyte senescence [100]. Therefore, it is suggested that OPN may be involved in OA processes and is a promising therapeutic agent in precision treatment of OA in the future [101]. 

Systemic lupus erythematosus (SLE) is a chronic connective tissue disease that affects nearly all of the important organs [102]. OPN is a biomarker and indicates bad prognosis of SLE. Circulating OPN levels are related to anti-dsDNA autoantibodies, subclinical atherosclerosis associated with SLE, and lupus nephritis [103,104,105].

Kon et al. demonstrated both in vivo and in vitro that there was a novel motif LRSKSRSFQVSDEQY in the *C*-OPN of MMP-3/7-cleaved mouse OPN, which bound to α9β1 integrin and was involved in the development of anti-type II collagen antibody-induced arthritis (CAIA) [106]. Dai et al. found that OPN and integrin β3 were upregulated in the infrapatellar fat pad (IPFP) and calcified cartilage in OA mice as well as in humans and that IPFP-derived OPN contributed to cartilage degeneration, subchondral bone remodeling and IPFP fibrosis via OPN–integrin β3 signaling. Their study also showed that intra-IPFP injection of RGD-nanogel/siRNA Cd61, which specifically targeted the IPFP cells expressing OPN receptors, effectively reduced the expression of integrin β3 and attenuated OA progression in mice [107]. Luo, W. et al. also reported that OPN, CD44 and HA synthase 1 (HAS1) were highly expressed in OA cartilage and chondrocytes, and OPN upregulated the expression of HAS1 and increased the anabolism of the synthesis of ECM components such as hyaluronic acid (HA) in cartilage through CD44 protein expression in OA mice, thereby inhibiting OA progression. Moreover, intra-articular injection of OPN in mice with OA significantly inhibits OA progression [101].

### 5.6. Nervous System Diseases

OPN is involved in the aging process of many systems, including central nervous system aging [108]. Alzheimer’s disease (AD) is characterized by synaptic loss, which may be caused by dysfunction of microglial phagocytosis and complement activation [109]. OPN promotes a proinflammatory and neurotoxic microglial phenotype via interaction with its integrin receptor αVβ3 and inhibits amyloid beta (Aβ) plaque compaction and clearance via suppressing the TREM2/TAM–lysosomal phagocytic pathway. Genetic deletion or antibody blockade of OPN reduces microglial inflammation, Aβ plaque pathology and neurotic dystrophy and improves cognitive function in AD. Increased microglial OPN production correlated positively with dementia severity and AD neuropathology in human brain tissue [18]. It has been shown that plasma OPN can be a biomarker of AD and vascular cognitive impairment [110]. Additionally, levels of OPN increase in the cerebrospinal fluid of patients with AD and are correlated with cognitive decline [111].

Parkinson’s disease (PD) is a clinical syndrome that represents a fast-growing neurodegenerative condition [112]. OPN increases in body fluids of PD patients. Higher serum levels of OPN are correlated with more severe motor symptoms, and higher CSF levels are positively associated with concomitant dementia and negatively related to dopaminergic treatment [113]. OPN is expressed in neurons, inducing mitochondrial dysfunction in human astrocytes [114]. However, different conclusions are reached in some studies. Reduced expression of OPN is observed in surviving dopaminergic neurons of PD patients. OPN protects dopaminergic cells from 1-methyl-4-phenylpyridinium toxicity and increases glial-derived neurotrophic factor and brain-derived neurotrophic factor levels, indicating that OPN is a double-edged sword in PD [115].

Multiple sclerosis (MS) is an autoimmune disorder of the central nervous system (CNS). Inflammation is a key factor in all MS stages [116]. Patients with relapsing–remitting MS had higher cerebral spinal fluid (CSF) levels of OPN and other inflammatory cytokines and adipokines [116,117]. Therefore, OPN may be a useful biomarker predicting disease activity in MS patients [117]. OPN is highly expressed within B cell aggregation in MS brain tissue, suggesting its role in B cell pathology. It is found that rOPN downregulates the co-stimulatory molecules CD80 and CD86 on B cells and reduces the production of IL-6 by B cells in vitro. However, rOPN also increases the tendency of B cells to form homotypic cell aggregation. OPN has opposing effects on B cell activation and aggregation, which may have an impact on the pathogenesis of B cell-mediated diseases [42]. In addition, OPN expression is elevated in DCs both in the periphery and in the central nervous system in experimental autoimmune encephalomyelitis (EAE), a mouse model of MS, as well as MS patients. There is also increased expression of the OPN receptors CD44, β3 and αv on T cells in MS patients. CD4+ T cells from MS patients produce significantly higher amounts of IL-17 when they are stimulated with OPN [118]. CSF levels of OPN are positively correlated with the proinflammatory cytokines IL-2 and IL-6 and negatively correlated with the anti-inflammatory molecule IL-1 receptor antagonist (IL-1ra) [117]. DC-produced OPN is associated with the production of IL-17 in both EAE and MS. Murugaiyan et al. found that OPN induced IL-17 production by CD4+ T cells via the β3 integrin receptor in MS patients while inhibiting IL-10 production via the CD44 receptor in EAE [118]. Moreover, OPN expression also increases in inflammatory cells and some neurons and blood vessels in the spinal cord, which has correlation with the severity of MS, inflammatory cell density and IL-17A expression [119].

Anti-OPN treatment is effective in reducing clinical severity of EAE by reducing IL-17 production [118]. Further, anti-α4β1 (VLA-4) antibody shows neuroprotective effects by reducing OPN expression and inflammation and increasing oligodendrocyte density [119].

### 5.7. Other Diseases

#### 5.7.1. Oral Diseases

OPN is involved in the pathogenesis of pulpitis and may be a potential therapeutic target. It is produced by dental pulp cells and may be linked to the calcification process of the pulp stone matrix [120]. Pulpitis is associated with increased expression of TLR2, TLR4 and OPN in the dental pulp. The researchers also found that TLR2 and TLR4 expression is positively correlated with OPN expression, suggesting a possible role of OPN in TLR-mediated inflammation [121].

Although OPN is known for its proinflammatory properties, it surprisingly serves as a protective agent against inflammation and bone damage in a mouse model of endodontic infection, indicating a potential therapeutic application in treating polymicrobial infections [122]. OPN enhances the cell surface expression of C–X–C motif chemokine receptors 2 (CXCR2) on bone marrow neutrophils in a way that depends on αv integrin, and it inhibits the internalization of CXCR2 when there is no ligand present. This process amplifies the capacity of these cells to move towards infection sites in response to CXCR2 ligands [123].

Isoforms of OPN could be potential biomarkers and therapeutic targets for periodontitis [124]. Treatment with Ixeris dentata (IXD), *Lactobacillus gasseri* media (LGM) or a combination of both on periodontitis in a mouse model inhibits alveolar bone loss and increases the expression of osteogenic factors, including OPN [125].

#### 5.7.2. Eye Diseases

The cornea, a specialized tissue that is transparent and lacks blood vessels, relies on a well-structured ECM to bend light and sustain its function. When damaged, keratocytes (also known as corneal fibroblasts) are activated and myofibroblasts are produced, resulting in the development of new blood vessels, scar formation and the clouding of the cornea [126]. Typical responses to injury in the corneal stroma include fibrosis and neovascularization [127]. The absence of OPN slows down the healing process of a cut injury in the corneal stroma. The deficiency of *OPN* inhibited the injury-induced increase in α-smooth muscle actin and the expression of fibrogenic genes. Further studies in cell culture reveal that ocular fibroblasts (sourced from the eyeshell of newborn mice) lacking OPN expressed fewer fibrogenic genes compared to those in normal cells [128]. Loss of OPN in ocular fibroblasts also suppresses expression of angiogenic cytokines [129]. In addition, the expression of OPN in the cornea changes dynamically during development, wound healing and diseases, which is related to corneal transparency and function. In addition, OPN participates in TGFβ-induced EMT in corneal epithelial cells, promoting corneal fibrosis and inflammation. It inhibits TGFβ signaling in corneal stromal cells, thus protecting corneal transparency and tensile strength, and affects the integrity and function of the corneal endothelial layer. The modulation of their functions could be a novel strategy to improve the outcome of corneal wound healing [32].

OPN may be a potential biomarker for retinal injury. In retinal degeneration (RD) mouse models, expression of OPN in RD retinas is increased and co-localized with microglial cells in the outer nuclear layer, outer plexiform layer and subretinal space [130].

#### 5.7.3. Allergic Diseases

In allergic contact dermatitis (ACD), OPN is abundantly expressed by both effector T cells and keratinocytes in lesions [131]. OPN supports DC migration and IL-12 expression and is secreted by T effector cells and keratinocytes, enhancing Th1-mediated allergy and supporting disease chronification [132]. Patients with acute ACD have a significantly higher percentage of iOPN-producing CD4+ T lymphocytes than healthy controls, which persists during remission [133]. Furthermore, acute disseminated ACD is characterized by elevated serum concentrations of OPN, with levels depending on ACD severity [134]. This indicates that OPN plays a role in the elicitation phase of ACD and could be used as an indicator of disease activity, with iOPN-producing T cells possibly participating in the effector phase of ACD [133,134]. The possibility of inhibiting OPN activity may provide a new therapeutic perspective for severe forms of this troublesome skin disease. Additionally, OPN may substitute for missing IFN-γ secretion in T effector cells because low-IFN-γ-producing T cell clones secrete high levels of OPN, and OPN downregulates their IL-4 expression. Moreover, IFN-γ from T effector cells enhances OPN in ACD by inducing OPN in keratinocytes, which in turn polarizes DCs and attracts inflammatory cells [131].

In allergic asthma, increased levels of OPN are found in bronchoalveolar lavage fluid (BALF). In *OPN*-deficient mice, higher levels of markers related to tissue injury and a higher bacterial burden in BALF and lung tissue are detected, suggesting the protective roles of OPN in asthma [135]. OPN can induce protective antigenic tolerance in mediastinal lymph nodes by inducing IFN-β-producing pDCs as well as regulatory T (Treg) cells, preventing patients from allergic airway inflammation [136].

#### 5.7.4. Skin Diseases

OPN is expressed in psoriasis lesions and enhances autoimmunity. Concentrations of OPN in the serum of psoriatic patients are higher than that in healthy controls [137]. In a psoriasis mouse model, *OPN* deficiency reduces IL-17 expression of inflammatory T cells, alleviating ear swelling and skin inflammation [138].

In pemphigus vulgaris patients, serum OPN levels are elevated [139].

#### 5.7.5. COVID-19

Plasma OPN levels are higher in COVID-19 patients compared with healthy controls. Persistently increased concentrations of OPN in serum are related to an unfavorable outcome in critically ill patients [140]. Thus, COVID-19 patients with critical illness demonstrate higher OPN levels than non-critically ill patients [141]. OPN is associated with adverse clinical outcomes and disease mortality in COVID-19 [142]. Moreover, a high OPN level increases the odds of mechanical ventilation requirement [143].

## 6. Conclusions

OPN, a natural protein in breast milk and infant formulas, is a multifunctional protein with immunomodulatory properties implicated in numerous diseases. It plays a crucial role in immune regulation, bone metabolism, inflammation, cell growth, migration and tumorigenesis. OPN also serves as a valuable biomarker for disease diagnosis, progression assessment and prognosis evaluation. Despite significant advances in understanding OPN’s immunomodulatory effects, further studies on the precise molecular mechanisms are still needed. It is also worth noting that OPN could serve as a potential therapeutic target for diseases prevention and treatment, which encourages us to deepen our understanding of the pathogenesis of the diseases.

## Figures and Tables

**Figure 1 nutrients-16-00312-f001:**
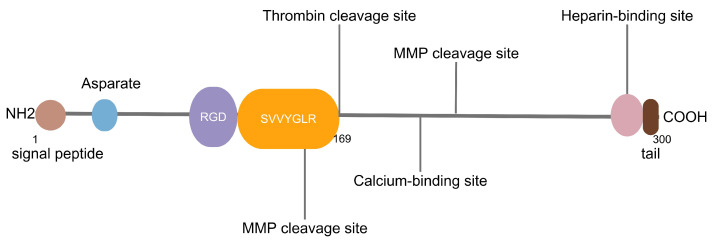
RGD: Arg-Gly-Asp, MMP: matrix metalloproteinase. The RGD motif and SVVYGLR domain are shown in purple and yellow, respectively. The known binding sites for heparin and calcium as well as cleavage sites for thrombin and MMPs are shown by lines.

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
