# Peer review of "Immunoregulatory Roles of Osteopontin in Diseases"

_nutrients, 2024, doi:10.3390/nu16020312_

Round 1
Reviewer 1 Report
Comments and Suggestions for Authors
Review Report
Immunoregulatory Roles of Osteopontin in Diseases
Lebei Wang, Xiaoyin Niu and Xiaoyin Niu
Comments and Suggestions for Authors
Osteopontin (OPN) as a growth regulatory protein, multifunctional cytokine, and adhesion molecule, is involved in the pathological processes of many diseases.
2. Function of OPN - I suggest summarizing the functions of OPN in table 1, it will be clearer.
3. Regulatory Effect of OPN on immune cells – the text contains a lot of information they could be conveniently included in the table 2. The effect of OPN can be divided into a stimulatory and inhibitory, the influence on various cell types, receptor and costimulatory molecules expression, cytokines production.
The details of OPN mechanisms can be in paragraph 4 (Immunoregulatory roles of OPN in diseases). It is optimal to present them in the form of a picture and an explanation below the picture.
4. Immunoregulatory roles of OPN in diseases - Divide diseases by creating two basic groups, chronic inflammatory and infectious diseases. Also add to this paragraph other diseases, e.g. autoimmune disorder, respiratory, cardiovascular, immune mediated, and virus diseases.
4.1. Digestive system diseases – celiac disease, irritable bowel syndrome, ulcerative colitis, Crohn´s, are not mentioned.
“The recruited macrophages were divided into two subsets, each with unique activation states.” Specify groups of macrophages in the text.
4.2. Urinary system diseases – diseases of the urinary system can be divided into inflammatory and noninflammatory. Expand the group, for example by bladder infections, etc.
“OPN secretion is triggered by proinflammatory cytokines, and OPN directly influences stromal cells to promote the production of proinflammatory mRNAs.” Proinflammatory mRNAs replace with mRNA for proinflammatory cytokines.
4.3. Hematological and hematopoietic system diseases - the description of animal models (e.g., lymphoma in mice) can be given in paragraph 2 (Function of OPN) or in a new paragraph OPN and in vivo/in vitro models.
Clinical works are missing, e.g., enhanced production of OPN in multiple myeloma patients, OPN´s relationships with prognostic factors and survival in diffuse large B cell lymphoma, a type of non-Hodgkin lymphoma.
4.4. Endocrine and metabolic diseases – mice model of hypertension to move to new paragraph in vivo/in vitro model.
Insert text and citations of clinical investigation, e.g., major role of OPN in the pathogenesis of hypertension, OPN as novel cardiac specific biomarkers, circulating OPN as independent risk factor for left ventricular hypertrophy and left ventricular diastolic dysfunction in essential hypersensitive patients. OPN is a robust predictor of incipient diabetic nephropathy, may play an important role in the development of insulin resistance by increasing inflammation and the recruitment of macrophages in adipose tissue.
4.5. Rheumatic diseases – move RA mouse model to new paragraph in vivo/in vitro model. On this place focus on soluble OPN levels, RA and osteoarthritis in patients, OPN as agent for promoting joint degradation in RA immunopathogenesis, point to autoimmune disorders (RA, SLE, multiple sclerosis, type 1 diabetes, inflammatory bowel disease, etc.).
4.6. Nervous system diseases – OPN in neurodegenerative diseases AD, SM, expand e.g., by Parkinson´s disease, role of OPN in microglia. Recent studies have found that OPN is widely involved in the aging processes – should be mentioned in the text.
4.7. Other diseases – insert to the text e.g., retinal degeneration, bone diseases, consider the creation of a separate paragraph OPN and allergic diseases (allergic contact dermatitis, allergic asthma).
I suggest adding the text: OPN as a biomarker for COVID severity, OPN and skin diseases, OPN and prognostic biomarker in critically ill patients, and creating a separate paragraph: the role of OPN in inflammatory process.
There are a lot of abbreviations throughout the text, definition of some is missing. After editing the text and incorporating comments, the work will be an enriching contribution to the field.
Reviewer 2 Report
Comments and Suggestions for Authors
In the present manuscript by Wang and Niu, the authors have concisely reviewed the literature on the immunoregulatory role of Osteopontin in various biological processes.
The manuscript is well-written. I have just a few comments.
Line 14: Shorten this phrase somewhat. “Here, we provide a systematic overview of OPN and its immunoregulatory roles in various diseases and highlighting possible approaches for the future development of OPN-based therapies”. Similar, lines 30-32.
Line 20-22 Remove ‘plays a crucial role in the immune response. It”. This will become clear later in the manuscript.
Page 22-23 .. is detected naturally in bone, kidney and lung and is furthermore expressed by various immune cells, including…
Line 25 “acts as a modulator of the immune response”: an rather vague statement; be more specific as to the modulating effect.
Line 34 “phosphorylated glycophosphoprotein” is a pleonasm, just state “phosphorylated glycoprotein”
Line 58 “derived from multiple cell types”, already stated above, please remove
Lines 60-62 “It also plays important roles in several physiological and pathological processes, including inflammation, tissue repair, and bone remodeling, making it 61 a promising target for therapeutic interventions” Please remove this general statement.
Line 83 Replace “in macrophage cells” “by macrophages “
Line 83 “The protein has been implicated in multiple functions of macrophages”. Rather vague. How? Be more specific. Also reviewed in reference 37.
Line 90 Mention that this has been observed in the context of multiple sclerosis or remove; because mentioned in lines 304-306.
Line 103 “n Th cells, the interaction between OPN and 103 CD44 triggers the hypomethylation of IFN-γ and IL17A genes, thereby promoting the differentiation of Th1 and Th17 cells” The reference to this statement is not clear, neither is the statement itself: it is somewhat unusual that one factor induces the differentiation of two different Th cell types
Line 122 “OPN has recently been described as a good biomarker of NASH in patient serum”. This statement is either erroneous or reference 47 is not correct.
Line 154 Which pro-inflammatory cytokines and how? Be more specific.
Comments on the Quality of English Language
The quality of the English language is OK. Some small errors could be corrected.
Round 2
Reviewer 1 Report
Comments and Suggestions for Authors
the manuscript, revised version, Title: Immunoregulatory Roles of Osteopontin in Diseases; Authors: Lebei Wang, Xiaoyin Niu has been sufficiently improved